# Genetic Diversity of Norovirus in Children with Acute Gastroenteritis in Southwest Nigeria, 2015–2017

**DOI:** 10.3390/v15030644

**Published:** 2023-02-28

**Authors:** Kafayat O. Arowolo, Christianah I. Ayolabi, Isaac A. Adeleye, Bruna A. Lapinski, Jucelia S. Santos, Sonia M. Raboni

**Affiliations:** 1Department of Microbiology, University of Lagos, Lagos 100213, Nigeria; 2Virology Laboratory, Infectious Diseases Division, Complexo Hospital de Clinicas, Universidade Federal do Paraná, Curitiba 80060-900, PR, Brazil

**Keywords:** acute gastroenteritis, norovirus, recombinant strain, genetic diversity

## Abstract

Norovirus (NoV) is a leading cause of viral gastroenteritis globally, especially in children below five years. Epidemiological studies on the diversity of NoV in middle- and low-income countries, including Nigeria, are limited. This study aimed to determine the genetic diversity of NoV in children below five years with acute gastroenteritis at three hospitals in Ogun State, Nigeria. A total of 331 fecal samples were collected from February 2015 to April 2017, while 175 were randomly selected and analyzed using RT-PCR, partial sequencing and phylogenetic analyses of both the polymerase (RdRp) and capsid (VP1) genes. NoV was detected in 5.1% (9/175; RdRp) and 2.3% (4/175; VP1) of samples, with 55.6% (5/9) co-infection with other enteric viruses. A diverse genotype distribution was identified, and GII.P4 was the dominant RdRp genotype detected (66.7%), with two genetic clusters, followed by GII.P31 (22.2%). The rare GII.P30 genotype (11.1%) was detected at a low rate for the first time in Nigeria. Based on the VP1 gene, GII.4 was the dominant genotype (75%), with two variants, Sydney 2012 and possibly New Orleans 2009, co-circulating during the study. Interestingly, both intergenotypic, GII.12(P4) and GII.4 New Orleans(P31), and intra-genotypic, GII.4 Sydney(P4) and GII.4 New Orleans(P4), putative recombinant strains were observed. This finding suggests the first likely report of GII.4 New Orleans(P31) in Nigeria. In addition, GII.12(P4) was first described in Africa and globally in this study, to the best of our knowledge. This study provided insights into the genetic diversity of NoV circulating in Nigeria, which would be useful for ongoing and future vaccine design and monitoring of emerging genotypes and recombinant strains.

## 1. Introduction

In low-income countries, particularly sub-Saharan Africa, acute gastroenteritis (AGE) is responsible for high morbidity and mortality in children below five years due to poor health care systems, poor hygiene practices and less access to clean portable water [1]. Enteric viruses, especially group A rotavirus (RVA) and norovirus (NoV), are significant etiologies associated with AGE [2]. With the adoption of rotavirus vaccines in the expanded program on immunization (EPI) of various countries [3], as well as advances in molecular testing, NoV has been considered a leading cause of viral gastroenteritis in children worldwide [4,5].

NoVs, belonging to the family Caliciviridae, are diverse, naked, icosahedral, positive-sense, single-stranded RNA viruses with a 7.5 kb genome containing three open reading frames (ORFs): ORF1, ORF2, and ORF3 [6]. The ORF1 encodes six non-structural proteins, including RNA-dependent RNA polymerase (RdRp), while ORF2 and ORF3 code for the major and minor capsid proteins VP1 and VP2, respectively [7]. Based on the VP1 amino acid sequence, NoV is categorized into ten genogroups (GI-GX), with GI, GII, GIV, GVIII and GIX mainly responsible for human infections [8]. Of these genogroups, GII is the most commonly detected in clinical surveillance studies worldwide [9]. Genogroups are sub-divided into 48 capsid genotypes and 60 P-types based on complete VP1 amino acid sequence and partial RdRp region nucleotide sequences, respectively. Genotypes are further sub-typed into variants based on >5% VP1 amino acid sequence variation between strains [8,10].

Worldwide, NoV GII.4 genotype is predominantly responsible for sporadic and outbreaks of NoV AGE. Six pandemic variants (US 95/96, Farmington Hills 2002, Hunter 2004, Den Haag 2006, New Orleans 2009, Sydney 2012), which emerged every two to three years over the last two decades, have been reported [11]. Recently, a new variant, GII.4 Hong Kong, circulating at a low level was reported in China [12]. Broad receptor specificity, high mutation rates and short-term immunity compared to other strains have been suggested for GII.4 dominance [13]. In addition, studies have shown that susceptibility to different NoV genotypes depends on individuals’ expression of histoblood group antigens (HBGA) [14].

Recombination is a common mechanism in NoV that increases its diversity and evolution. It mainly occurs at the ORF1/ORF2 junction and results in a recombinant strain clustering into two different genotypes when sequenced using two separate regions of the genome (VP1 and RdRp) [15]. This inter-genotypic recombination, coupled with antigenic drift, contributes to the frequent emergence of new NoV strains that can escape human herd immunity. Hence, a dual-typing classification system was proposed to account for all recombinant strains [8,10]. Currently, there are several NoV vaccine candidates in the preclinical and clinical phases of development [16].

While extensive studies have been carried out on the prevalence and diversity of NoV in high- and upper-middle-income countries, only a few studies have been reported in middle- and low-income countries where high mortality rates exist [17]. The pooled prevalence of NoV in children below five years with AGE between 2015 and 2021 was 19.25% (95% CI: 14.4, 23.5) in Africa, with GII.4, GII.17 and GI.3 as predominant genotypes [18]. In Nigeria, existing studies reported NoV prevalence ranging from 3.6% to 37.3% [19,20,21,22,23,24,25], with data on genotype distribution from solely two studies [21,23]. However, only Japhet et al. reported combined RdRp and VP1 typing [23]. Hence, there needs to be more information regarding genetic diversity and recombinant strains of NoV in Nigeria. This data is imperative for a successful vaccine design that would be effective against numerous NoV strains circulating amongst children in developing countries where a significant impact is expected. Therefore, this study aimed to determine the genetic diversity of NoV in children below five years old with acute gastroenteritis in Southwest Nigeria.

## 2. Materials and Methods

### 2.1. Ethical Approval

The Institutional Ethical Committee of the Federal Medical Centre, Abeokuta (FMCA/470/HREC/09/2015), State Hospital, Ijebu-Ode (RER/AP/VOL/113) and State Hospital, Ota (2265/02), Ogun State Nigeria approved the study. The parents/guardians of participants gave their informed written consent before enrollment into the study.

### 2.2. Sample Collection

This is a cross-sectional hospital-based study conducted from February 2015 to April 2017 at the three hospitals mentioned above, located in each of the three senatorial districts, simultaneously, across Ogun State, Nigeria. Stool samples were collected from 331 children aged ≤five years who presented at the pediatric clinics with an AGE of ≤seven days’ duration. AGE is characterized by diarrhea- the passage of three or more loose or liquid stools within 24 h. The parents/guardians provided socio-demographic and clinical information by completing a standardized questionnaire for each subject. Of the 331 stool samples, 175 were selected randomly for analysis.

### 2.3. Detection of NoV 

The stool samples from participants were processed, and viral nucleic acids were extracted as previously described [26]. NoV was detected using reverse-transcription polymerase chain reaction (RT-PCR) to amplify the partial RdRp gene (region A) and VP1 gene (region D) using specific primers [27,28]. Concisely, complementary DNA (cDNA) was synthesized with 10 µL extracted nucleic acid using random primers and Superscript III Reverse Transcriptase Kit (Invitrogen, Carlsbad, CA, USA) according to the manufacturer’s instructions. Region A was amplified using 2.5 µL of cDNA in a 25 µL PCR reaction mix containing 0.5 µL dNTPs (10 mM each), 2.5 µL PCR 10× buffer, 0.75 µL MgCl2 (50 mM), 0.5 µL primer set (10 pmol/µL), 0.125 µL Platinum Taq DNA Polymerase (200 U/µL; Invitrogen) and 17.625 µL RNase-free water. The PCR conditions in a Veriti 96-well thermal cycler (Applied Biosystems Inc., CA, USA) were as follows: 94 °C for 3 min, 40 cycles of 94 °C for 30 s, 50 °C for 30 s, 72 °C for 30 s and 72 °C for 2 min. In addition, region D was amplified with the same reaction mix using a genogroup II specific primer set (10 pmol/µL) under these PCR conditions: 94 °C for 3 min, 35 cycles of 94 °C for 45 s, 45 °C for 45 s and 72 °C for 60 s. All PCR products were resolved by electrophoresis on a 1.5% agarose gel stained with ethidium bromide and observed using a UV transilluminator. Specific-sized bands were analyzed using the E-Capt program (Vilber Lourmat Deutschland, GmbH, Germany). Detection of group A rotavirus [RVA], human astrovirus (HAstV) and human adenovirus (HAdV) was also performed concurrently using RT-PCR and PCR, as described previously [20].

### 2.4. Nucleotide Sequencing and Phylogenetic Analysis

The positive PCR amplicons for the partial regions A and D were purified using a QIAquick Gel Extraction Kit (QIAGEN, Germany) according to the manufacturer’s instructions and subjected to direct Sanger sequencing in both directions with the same specific PCR primers using a Big Dye^®^ Terminator v.3.1 Kit and an ABI 3130 automated sequencer (Applied Biosystems, Foster City, CA, USA). The sequencing chromatograms were assembled and edited using the SeqMan tool in DNASTAR Lasergene 7.0.0 software (Dnastar, Inc.m Madison, WI, USA). NoV genotypes/variants were assigned using the online NoV typing tool version 2.0 [29] and subsequently inferred by phylogenetic analyses. Generated nucleotide sequences were compared with prototype/reference strains of human NoV retrieved from the GenBank database, and multiple sequence alignments were performed using ClustalW. The phylogenetic trees of both regions were constructed by the maximum likelihood (ML) method with IQ-TREE v1.6.12 through the online server (http://iqtree.cibiv.univie.ac.at/230108) [30], using an automatic substitution model (ModelFinder), ultrafast bootstrap (UFBoot) [31] and SH-aLRT test (1000 replicates). Sequence divergence was calculated using mean pairwise distances. Phylogenetic trees were drawn in FigTree v1.4.4 (http://tree.bio.ed.ac.uk/software/figtree/). The NoV strains that clustered with different genotypes based on phylogenetic analysis of both regions were considered as recombinant strains. The nucleotide sequences obtained in this study were deposited in the GenBank database with accession numbers MN505792- MN505800 and MN508789-MN508792 for RdRp and VP1 gene sequences, respectively.

## 3. Results

Of the 175 randomly selected stool samples collected from children with AGE, NoV was detected in 9 (5.1%) and 4 (2.3%) samples based on partial RdRp region A and VP1 region D, respectively. These NoV-positive samples were detected in the State Hospital, Ijebu-Ode, and belonged to GII. Co-infection between NoV and other tested enteric viruses was observed in 55.6% (5/9) of positive samples, while 44.4% (4/9) had mono-infection (Figure 1). The proportion of co-infection was equal (40%, 2/5) for NoV with HAstV and NoV with HAdV, while mixed infection of NoV with RVA and HAdV existed in the lowest frequency (20%, 1/5). The highest proportion of NoV infection (6/59; 10.2%) was detected in children 0–12 months of age (Appendix A).

Concerning the RdRp region A, all nine NoV-positive samples were sequenced and three different genotypes: GII.P4, GII.P30, and GII.P31, were identified in children with AGE. The GII.P4 was the predominant genotype with a frequency of 66.7% (6/9), followed by GII.P31 (22.2%; 2/9) (Figure 2A). As shown in Figure 1, GII.P4 had the highest proportion (50%; 2/4) of mono-infection. Based on the VP1 region D, the four NoV-positive samples were sequenced and two genotypes: GII.4 and GII.12, were assigned. The GII.4 genotype was predominant with 75% (3/4) prevalence (Figure 2B). Of the three identified GII.4 genotypes, only one was assigned as Sydney 2012 variant while two could not be assigned to any variants by the NoV typing tool. As shown in Figure 2C,D, NoV genotypes were detected in the second calendar year of the study period (2016–2017), with 66.7% (6/9) and 75% (3/4) of RdRp-based and VP1-based genotypes detected in 2017, respectively. 

Phylogenetic analysis of the partial RdRp region revealed that GII.P4 strains identified in this study formed two distinct clusters. Cluster A, comprising five strains, 96%–100% nucleotide (nt) identical to each other, was closely related to GII.P4 strains from Nigeria (KU173783–KU173786) previously detected in 2012 and 2013 (97% nt identity). Cluster B comprised one strain similar to the GII.P4 strain from Belgium (EU794898) detected in 2006 (99% nt identity). However, it showed a lower similarity with previously detected Nigerian strains (92%–93% nt identity). Cluster A strains were distantly related to the cluster B strain with 91%–93% identity (Figure 3A). 

The GII.P31 strains from this study were 95% similar to each other and clustered alongside previous strains detected in Nigeria (KU173780, KU173781, and KU173782) with 92%–98% nt identity. Of the GII.P31 strains, sequence MN505799 was closely related to strains from Gabon (MW513406 and MW513408) detected in 2018, showing 100% nt sequence identity. However, the MN505797 sequence was closely related only to the Japanese strain with 96% nt identity (Figure 3A). 

The only GII.P30 strain clustered alongside the prototype strain (AY134748) with 87% nt identity and showed a higher similarity (92% nt sequence identity) with the Cameroonian strain (MH608285) detected in 2014. However, it showed 88%–91% nt sequence identity with other African reference strains (Figure 3A).

The phylogenetic analysis of the partial VP1 region of the study strains is shown in Figure 3B. Of the three GII.4 identified strains, one belonged to the Sydney 2012 variant, forming a distinct cluster with reference strains detected in different countries. This strain showed higher similarity with MF401681 and LC175468 detected in 2014 and 2016, respectively (94% nt identity). The other two GII.4 strains, unassigned by the NoV typing tool, were 100% identical and formed a cluster with New Orleans 2009 variant reference strains, with 100% nt identities. However, the bootstrap support value for this clade is poorly supported (42%) in the phylogenetic tree. The only GII.12 strain clustered alongside the prototype strain detected in the UK (AJ277618) with 83.0% nt sequence identity and showed higher similarity with the reference strain (MH608285) detected in Cameroon in 2014 (89% nt identity). 

The combined RdRp and VP1 genotypes (dual typing) of four strains were determined by phylogenetic analysis and resulted in four different recombinants, comprising two (50%) each intergenotypic and intragenotypic recombinant strains. The intergenotypic putative recombinant strains include GII.12(P4) and possibly GII.4 New Orleans(P31) while GII.4 Sydney(P4) and possibly GII.4 New Orleans(P4) are intragenotypic putative recombinants. There was also an increase in the proportion of recombinant strains from 2016 (25%) to 2017 (75%).

## 4. Discussion

Globally, viral gastroenteritis is still a public health issue, and NoV is significantly responsible for outbreaks and sporadic cases. The NoV detection rates of 5.1% and 2.3%, based on RdRp and VP1 regions, respectively, were lower than those reported in some earlier studies in Nigeria [21,22,23,24,25], and the pooled prevalence of 19.25% in Africa between 2015 and 2021 [18]. The observed discrepancy might be due to geographical location, sample size, study population, and varying diagnostic methods. This finding may suggest a possible decline of NoV-associated gastroenteritis in Nigeria since the last eight years before this study (2007–2014), as this study was conducted between 2015 and 2017. However, there is likely to be an increase in the NoV detection rate due to the recent introduction of RVA vaccination into the country’s EPI in August 2022, as was the case with some African countries that reported a higher detection rate after the implementation of RVA vaccines [32,33].

The dual infections of NoV with HAstV and NoV with HAdV observed in this study were in contrast to Japhet et al. [23], who reported neither of these combinations nor mixed infection in Osun State, Nigeria. However, dual NoV and HAdV infection and mixed NoV, RVA, and HAdV infection have been reported in Edo State, Nigeria [19], and Kenya [34]. Elsewhere, a study in Brazil [35] also reported dual NoV and HAstV infection. The number of viruses analyzed, sample size, and diagnostic methods may account for the disparity. NoV was detected exclusively in samples collected from the State Hospital, Ijebu-Ode, which suggests that there was an active shedding/transmission of this virus in children living in the senatorial district of the State where the Hospital is located. However, this finding might not reflect the senatorial district with the highest burden of NoV, as only participants who presented in the hospitals were included in the study.

The only NoV genogroup identified in this study was GII, which contrasts with previous reports in Nigeria [21,23,25] and other African countries [13,32,33,36] that observed both GI and GII genogroups. In line with this study, GII was reported as the only genogroup in Vietnam and Northern Italy [37,38]. Globally, GII is considered the predominant genogroup [18]. One of the RdRp- based genotypes identified in this study is GII.P31, which has been reported previously in Osun State, Nigeria [23], and elsewhere [13,39]. This genotype has been responsible for outbreaks in several countries [40]. Thus, attention should be paid to its future spread in Nigeria. Phylogenetic analysis revealed close sequence similarity between this study’s GII.P31 strains and those from different countries, indicating multiple introduction sources with diverse geographical clustering.

GII.P4 was the dominant RdRp-based genotype in this study, which is consistent with previous studies in Osun State, Nigeria [23] and other parts of the world [33,41]. Moreover, phylogenetic analysis showed that two genetic clusters of GII.P4 strains were co-circulating in Ogun State, with one (cluster A) genetically related to those reported previously in Osun State, Nigeria, between 2012 and 2013 [23]. These findings indicate the continuous spread of the GII.P4 genotype in southwestern Nigeria, which is not surprising, as both States share borders that facilitate the movement and interaction of people. Interestingly, the GII.P4 strain in cluster B is more related genetically to a Belgium strain reported in 2006. This suggests a possible accumulation of point mutations resulting in a significant divergence from the previous Nigerian strains and suggestive of NoV continuous evolution in Ogun State, hence the need for surveillance in the country. In addition, this finding might indicate that this strain was imported from Belgium to Nigeria. Furthermore, the GII.P4 strains in this study circulated predominantly in 2017. This observation is expected due to chance, as the NoV prevalence rate may fluctuate over time.

The rare GII.P30 genotype was detected for the first time in Nigeria in this study and circulated only in 2017 at a low rate. This indicates that it is an emerging genotype in Nigeria, which explains its non-occurrence in other studies in the country. However, it has been reported elsewhere in Africa [32,39]. The GII.P30 strain in this study clustered with those identified in other African countries, indicating that the genotype is circulating in the region. Moreover, it showed close sequence similarity with the GII.P30 strain from Cameroon, which suggests possible importation into the country due to their shared borderline. In addition, the GII.P30 prototype strain (Snow Mountain Virus; AY134748), a naturally occurring recombinant that caused a waterborne AGE outbreak in the USA in 1976 [42], is the evolutionary ancestor of these African strains, including the Nigerian strain. Hence, molecular surveillance of this genotype is crucial as it could be responsible for a future outbreak in the country.

Similar to the RdRp genotypes, GII.4 was the dominant genotype based on the VP1 region. This is consistent with previous studies in Nigeria [21,23] and elsewhere globally [18]. This finding further corroborates the predominant widespread distribution of the GII.4 genotype over other GII genotypes. Two different GII.4 variants, Sydney 2012 and possibly New Orleans 2009, were co-circulated in this study, similar to the findings of Osazuwa et al. [21] in the south region of Nigeria and elsewhere [43]. Japhet et al. [23] detected only the Sydney 2012 variant between 2012 and 2013 in a southwestern State of the country. This suggests the gradual replacement of Sydney 2012 by the New Orleans 2009 variant, causing sporadic AGE in the country’s southwestern part. However, it is unsure whether the New Orleans variant would entirely displace Sydney 2012 in future sporadic cases of AGE in the geographical area. This observed pattern could be due to the different time frames of various studies. Therefore, further confirmatory studies with whole genome sequencing are needed to properly assign variants.

A non-GII.4 genotype, GII.12, was identified at a low frequency in this study, which is comparable with a previous study conducted between 2018 and 2019 in Nigeria [21]. This indicates that the GII.12 genotype was already circulating as of 2017, predating its earlier report in the country. Hence, it is an emerging genotype in southwest Nigeria, which explains its non-occurrence in a previous study in Osun State [23]. Similarly, the GII.12 genotype had been reported at low frequencies in several continents [34]. The genetic relatedness of this study’s GII.12 to a Cameroonian strain indicates its importation link.

Recombinant strains were observed based on the combined phylogenetic analysis of RdRp and VP1 genes, with two intragenotypic GII.4 recombinants. Interestingly, this is the first report of GII.12(P4) intergenotypic putative recombinant strain on the African continent and globally, to the best of our knowledge. A GII.12[P16] recombinant has been reported from North and Central America, Europe, and Australia/Oceania [44]. In contrast, studies in Africa [13,39,45] reported P31 and P33 polymerases associated with GII.12. This indicates a relatively high tendency of recombination among GII.12 strains. The second intergenotypic recombinant detected in this study was GII.4 New Orleans(P31) strain. While GII.4(P31) recombinant was reported previously in Nigeria [23], the Norovirus typing tool indicates that the GII.4 was a Sydney 2012 variant. Thus, this study is likely the first report of GII.4 New Orleans(P31) putative recombinant in Nigeria. However, more studies are required to ascertain if this recombinant is circulating widely in Nigeria. The GII.P31 is considered an obligatory recombinant strain associated with several different VP1 genotypes, including GII.4, GII.3 and GII.12 [46], which corroborates this study. In addition, GII.4 Sydney(P4) and probably GII.4 New Orleans(P4) intra-genotypic recombinants were detected in this study, further showing the diversity of the GII.4 genotype by recombination. In contrast, only GII.4 Sydney(P4) has been previously reported at a high frequency in Nigeria [23], suggesting a common phenomenon in the country. Elsewhere in Europe [47], GII.4 Sydney(P4) and GII.4 New Orleans(P4) were reported. The observed increase in the proportion of recombinant strains from 2016 to 2017 suggests a circulation pattern, although limited recombinants were detected.

While dual-typing of RdRp and VP1 regions was used in this study to determine recombinant strains, further confirmatory recombination analyses were not performed as generated sequences did not cover the overlapping ORF1/ORF2 junction nor the ORF2/ORF3 junction. In addition, two separate amplification assays were used for the analysis, which could lead to false recombinant strains due to co-infections with different NoV genotypes present in samples. Furthermore, due to the relatively small sample size and low positive samples, the NoV diversity may not represent the heterogeneous strains circulating in Ogun State and, in particular, Nigeria. However, various circulating genotypes were identified within the selected samples. Lastly, a control group of children without AGE was not included in the study to better understand the role of NoV in AGE.

In conclusion, this study shows lower NoV prevalence and a diverse range of NoV genotypes based on dual-typing from Ogun State between February 2015 and April 2017, providing the first described rare GII.P30 in Nigeria. GII.P4 and GII.4 were the most dominant circulating genotypes, with two genetic clusters and variants co-circulating, respectively. In addition, two intergenotypic putative recombinant strains were identified, and to the best of our knowledge, this is the first GII.12(P4) report in Africa and globally. This data highlights the importance of dual-typing and the need to sequence large genomic regions to allow robust NoV diversity crucial for ongoing vaccine development. Hence, continuous extensive hospital/community-based surveillance across the country and Africa is necessary to guide the production of effective cross-protective candidate vaccines against circulating strains.

## Figures and Tables

**Figure 1 viruses-15-00644-f001:**
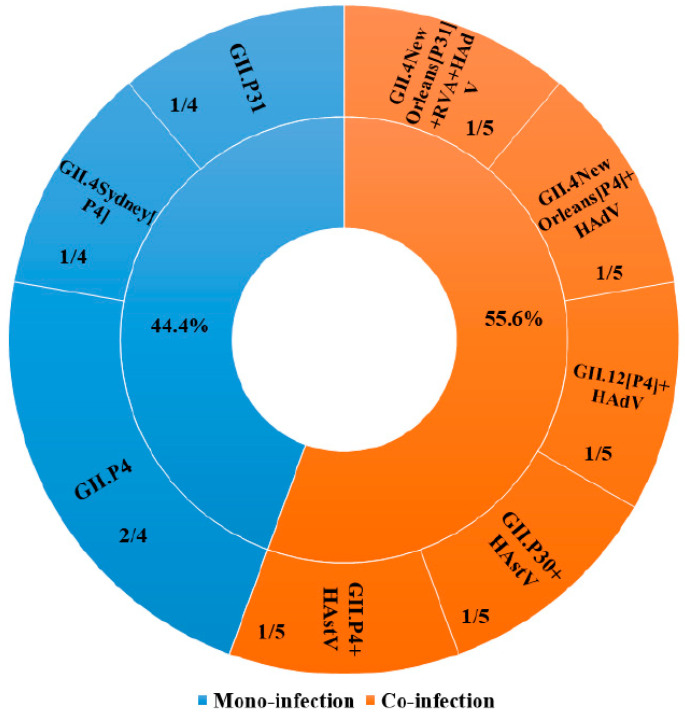
Co-infection patterns of NoV in children with acute gastroenteritis in Ogun State, Nigeria, from February 2015 to April 2017.

**Figure 2 viruses-15-00644-f002:**
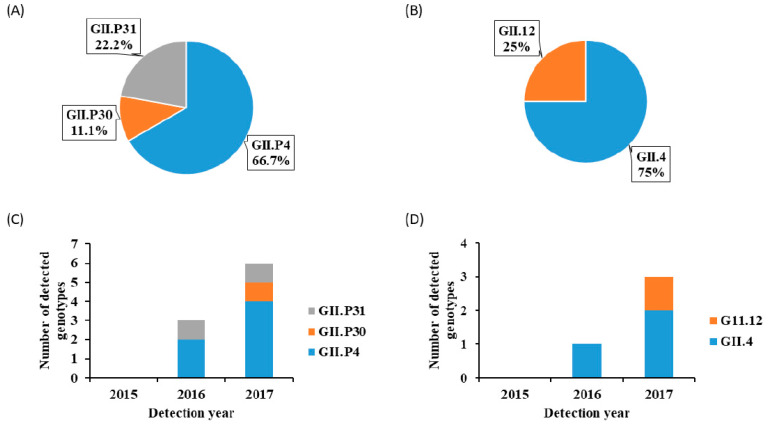
NoV distribution in children with acute gastroenteritis in Ogun State, Nigeria, from February 2015 to April 2017. (**A**) RdRp genotypes, (**B**) VP1 genotypes, (**C**) RdRp genotypes by year, and (**D**) VP1 genotypes by year.

**Figure 3 viruses-15-00644-f003:**
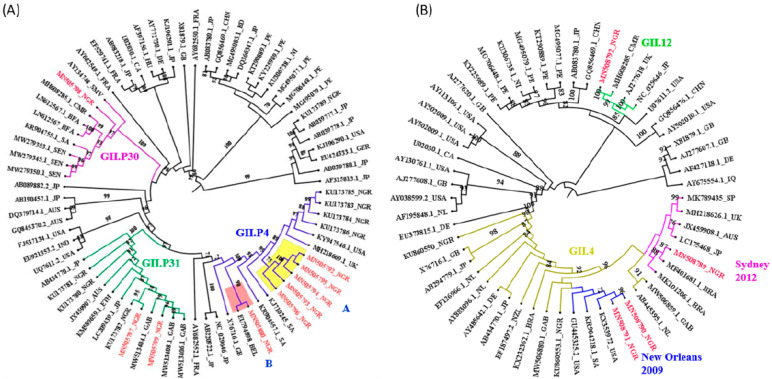
Maximum-likelihood phylogenetic trees of NoV strains identified in children with acute gastroenteritis in Ogun State, Nigeria, from February 2015 to April 2017. (**A**) Partial RdRp region A (**B**) Partial VP1 region D. Colored text indicates the study strains and reference strains obtained from the GenBank database are indicated by accession numbers (in black). Bootstrap support values > 70 (1000 replicates) are shown next to the branches.

## Data Availability

Sequences generated from this study are available in NCBI GenBank under accession numbers MN505792–MN505800 and MN508789–MN508792.

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
