# Peer review of "Genetic Diversity of Norovirus in Children with Acute Gastroenteritis in Southwest Nigeria, 2015–2017"

_viruses, 2023, doi:10.3390/v15030644_

Round 1
Reviewer 1 Report
The article by Arowolo et al. presents the novel finding of GII.4 New Orleans[P31] and GII.12[P4] strains of norovirus in Nigeria and hence, raises awareness towards monitoring of the emerging genotypes for necessary vaccine design. The authors have performed PCR for norovirus detection in patient fecal samples and then performed nucleotide sequencing and phylogenetic analysis. Here are some suggestions:
Line 102: Why were just 175 out of 331 samples chosen? This reduces the sample size as well as creates experimentally biased results.
Line 123: Although “described previously”, the methodology can be briefly summarized.
Line 146-153: It will be highly informative if these claims are supported by PCR analysis data/heatmap as a separate figure or supplementary figure. Maybe some statistical analysis can be done on the positive vs negative samples to show us the extend of norovirus infection in these patients. Additionally, the co-infection vs mono-infection and different types of co-infection data can also be represented in pie charts.
Please briefly mention why those certain strains (HAstV, RVA, HadV) were chosen to test co-infection.
As mentioned in discussion, the co-infection found in this study contrasts with the study by Japhet et al, 2012. Thus, the inference should be better supported with data and further experiments.
Figure 1: Can a better resolution figure be provided? Also, kindly expand the figure legend.
Can the infected vs non-infected also be divided on the basis of patient’s age?
Figure 2: Clusters A and B are not well-defined in the figure. Brackets/paranthesis/highlights can be added to identify them easily.
Line 318-319: Can this sentence about not having “a control group” be expanded on? Are there any future studies that can be done including samples from healthy subjects vs AGE subjects to extrapolate this study?
Author Response
Reviewer 1
The article by Arowolo et al. presents the novel finding of GII.4 New Orleans[P31] and GII.12[P4] strains of norovirus in Nigeria and hence, raises awareness towards monitoring of the emerging genotypes for necessary vaccine design. The authors have performed PCR for norovirus detection in patient fecal samples and then performed nucleotide sequencing and phylogenetic analysis. Here are some suggestions:
(1) Line 102: Why were just 175 out of 331 samples chosen? This reduces the sample size as well as creates experimentally biased results.
R - 175 samples were randomly selected for analysis due to limited funds, however, these samples cover the three hospitals located in each of the three senatorial districts of the State.
(2) Line 123: Although “described previously”, the methodology can be briefly summarized.
R - The methodology has been extensively discussed in a previous publication [20] which is not the main focus of this article to avoid lengthy and convoluted information. However, the molecular techniques used for the other viruses have been included in the sentence.
Detection of group A rotavirus [RVA], human astrovirus (HAstV) and human adenovirus (HAdV) was also performed concurrently using RT-PCR and PCR, as described previously [20].
(3) Line 146-153: It will be highly informative if these claims are supported by PCR analysis data/heatmap as a separate figure or supplementary figure. Maybe some statistical analysis can be done on the positive vs negative samples to show us the extend of norovirus infection in these patients. Additionally, the co-infection vs mono-infection and different types of co-infection data can also be represented in pie charts.
R - Based on this study design, the objective was not to determine the association between risk factors/demographics and the prevalence of NoV, hence statistical analysis was not included.
The coinfection and mono-infection pattern has been represented in Figure 1 and other Figures are renumbered.
Figure 1. Co-infection patterns of NoV in children with acute gastroenteritis in Ogun State, Nigeria, from February 2015 to April 2017.
(4) Please briefly mention why those certain strains (HAstV, RVA, HadV) were chosen to test co-infection.
RVA, HAstV and HAdV are the medically important enteric viruses responsible for acute gastroenteritis. This has been captured in reference [2].
(5) As mentioned in discussion, the co-infection found in this study contrasts with the study by Japhet et al, 2012. Thus, the inference should be better supported with data and further experiments.
R - I am not sure if the reviewer wants the co-infection observed by the Japhet et al study to be included in the statement. However, I noticed an error in the quoted reference [22] which I have updated to the correct one [23].
(6) Figure 1: Can a better resolution figure be provided? Also, kindly expand the figure legend.
R - The correction has been provided and all Figures have been renumbered.
Figure 2. NoV distribution in children with acute gastroenteritis in Ogun State, Nigeria, from February 2015 to April 2017. (A) RdRp genotypes, (B) VP1 genotypes, (C) RdRp genotypes by year, and (D) VP1 genotypes by year.
(7) Can the infected vs non-infected also be divided on the basis of patient’s age?
R - The comment has been captured in Figure S1. The sentence below has been written in the main text of the result section.
The highest proportion of NoV infection (6/59; 10.2%) was detected in children 0-12 months of age (Figure S1).
Figure S1. Age distribution of NoV infection in children with acute gastroenteritis in Ogun State, Nigeria, from February 2015 to April 2017.
(8) Figure 2: Clusters A and B are not well-defined in the figure. Brackets/paranthesis/highlights can be added to identify them easily.
R - The correction has been effected and all Figures have been renumbered.
(9) Line 318-319: Can this sentence about not having “a control group” be expanded on? Are there any future studies that can be done including samples from healthy subjects vs AGE subjects to extrapolate this study?
Lastly, a control group of children without AGE was not included in the study to better understand the role of NoV in AGE.
R - Further studies have been suggested in the concluding part of the manuscript to address all the limitations of this study.
Reviewer 2 Report
Dear Authors
Please note the following recommended edits to the manuscript;
1. Page 2 lines 62-64
Reference 13 was published in 2018 and describes norovirus infections in Botswana, Africa. Reference 12 was published in 2021 and describes detection of variant GII.4 Hong Kong in Asia and Europe. Reference 13 is not in any way directly related to the GII.4 Hong Kong variant, yet in page 2 line 64, reference 13 is cited as the text that "suggests" factors for the "dominance" of the GII4 Hong Kong variant.
Please select a correct reference or rephrase to eliminate incorrect interpretation of reference 13.
2. Page 4 line 150
It is not clear which norovirus strains were present as mixed co-infections or as single infections i.e. positive by region A or positive by region D
3. Page 4 line 157
Keep numbering pattern consistent i.e. 22.2% (2/9)
4. Page 4 Figure 1
Figure 1 is too small and not clear. The size of both the figure and associated text must be increased.
NB: the same comment applies to Figure 2 (Page 5).
Increase the size of figure 2 to become more clear and interpretable
5. Page 4 lines 154-160
The paragraph is not easily read with reference to figure 1. Indicate clearly in the text when Fig 1A is being discussed and then Fig 1B.
6. Page 4 line 162
This statement of "two years" is not accurate. The duration of February 2016 (the beginning of the second calendar year of the study) to April 2017 is not two years. It is 14 months
7. Page 6 lines 233-237
Comparisons with genogroup GI detected in other diversity study reports cannot be made since the screening assay in this investigation was specific for genogroup GII detection- cross reference line 116 in page 3
Comment: Confirmatory recombination analyses will verify calling the rare strains observed as "recombinants." In the meanwhile, consider referring to these noroviruses as "putative recombinants."
Author Response
Reviewer 2
Please note the following recommended edits to the manuscript;
(1) Page 2 lines 62-64
Reference 13 was published in 2018 and describes norovirus infections in Botswana, Africa. Reference 12 was published in 2021 and describes detection of variant GII.4 Hong Kong in Asia and Europe. Reference 13 is not in any way directly related to the GII.4 Hong Kong variant, yet in page 2 line 64, reference 13 is cited as the text that "suggests" factors for the "dominance" of the GII4 Hong Kong variant.
Please select a correct reference or rephrase to eliminate incorrect interpretation of reference 13.
The factors suggested for dominance were not referring to GII.4 Hong Kong but to GII.4 genotype mentioned at the start of the paragraph. Hence reference [13] is correct. The sentence has been updated to give a correct interpretation.
Broad receptor specificity, high mutation rates and short-term immunity compared to other strains have been suggested for GII.4 dominance [13].
(2) Page 4 line 150
It is not clear which norovirus strains were present as mixed co-infections or as single infections i.e. positive by region A or positive by region D
The correction has been represented in Figure 1 and other Figures are renumbered. The sentence below is mentioned in the second paragraph of the result section to reflect Figure 1.
As shown in Figure 1, GII.P4 had the highest proportion (50%; 2/4) of mono-infection.
(3) Page 4 line 157
Keep numbering pattern consistent i.e. 22.2% (2/9)
The correction has been effected.
The GII.P4 was the predominant genotype with a frequency of 66.7% (6/9), followed by GII.P31 (22.2%; 2/9).
(4) Page 4 Figure 1
Figure 1 is too small and not clear. The size of both the figure and associated text must be increased.
NB: the same comment applies to Figure 2 (Page 5).
Increase the size of figure 2 to become more clear and interpretable
The correction has been effected and all Figures have been renumbered.
(5) Page 4 lines 154-160
The paragraph is not easily read with reference to figure 1. Indicate clearly in the text when Fig 1A is being discussed and then Fig 1B.
The correction has been effected and all Figures have been renumbered.
(6) Page 4 line 162
This statement of "two years" is not accurate. The duration of February 2016 (the beginning of the second calendar year of the study) to April 2017 is not two years. It is 14 months
The Comment: Confirmatory recombination analyses will verify calling the rare strains observed as "recombinants." In the meanwhile, consider referring to these noroviruses as "putative recombinants."
As shown in Figures 2C and 2D, NoV genotypes were detected in the second calendar year of the study period (2016-2017), with 66.7% (6/9) and 75% (3/4) of RdRp-based and VP1-based genotypes detected in 2017, respectively.
(7) Page 6 lines 233-237
Comparisons with genogroup GI detected in other diversity study reports cannot be made since the screening assay in this investigation was specific for genogroup GII detection- cross reference line 116 in page 3
This study used both the polymerase (RdRp) gene (region A) and the capsid (VP1) gene (region D) for amplification and sequencing. The polymerase gene is a highly conserved region at the 5’ end of the NoV genome which is used for the identification of both genogroups G1 and GII strains. Hence, the reason for comparison with other studies that detected GI strains. The primer set targeting the VP1 region D was the only genogroup II-specific primer used in the study.
(8) Comment: Confirmatory recombination analyses will verify calling the rare strains observed as "recombinants." In the meanwhile, consider referring to these noroviruses as "putative recombinants."
We agree with the reviewer and the correction has been effected.